# Doob's Lagrangian:
# A Sample-Efficient Variational Approach to Transition Path Sampling

**Yuanqi Du** [* 1]   **Michael Plainer** [* 2]   **Rob Brekelmans** [* 3]   **Chenru Duan** [4]   **Frank Noe** [5 6 7]   **Carla P. Gomes** [1]
**Alan Apsuru-Guzik** [3 8]   **Kirill Neklyudov** [9 10]

## Abstract

Rare event sampling in dynamical systems is a fundamental problem arising in the natural sciences, which poses significant computational challenges due to an exponentially large space of trajectories. For settings where the dynamical system of interest follows a Brownian motion with known drift, the question of conditioning the process to reach a given endpoint or desired rare event is definitively answered by Doob's $h$-transform. However, the naive simulation of this transform is infeasible, as it requires sufficiently many forward trajectories to estimate rare event probabilities. In this work, we propose a variational formulation of Doob's $h$-transform — an optimization problem over trajectories between a given initial point and the desired ending point. To solve this optimization, we propose a simulation-free training objective with a model parameterization that imposes the desired boundary conditions by design. Our approach significantly reduces the search space over trajectories and avoids expensive trajectory simulation and inefficient importance sampling estimators which are required in existing methods. We demonstrate the ability of our method to find feasible transition paths on real-world molecular simulation and protein folding tasks.

## 1. Introduction

Conditioning a stochastic process to obey a particular endpoint distribution, satisfy desired terminal conditions, or observe a rare event is a problem with a long history (Schrödinger, 1932; Doob, 1957) and wide-ranging appli-

*Equal contribution  [1]Cornell University  [2]Technische Universität Berlin  [3]Vector Institute  [4]Massachusetts Institute of Technology  [5]Freie Universität Berlin  [6]Rice University  [7]Microsoft Research AI4Science  [8]University of Toronto  [9]Université de Montréal  [10]MILA Quebec AI Institute. Contact: yuanqidu@cs.cornell.edu , michael.plainer@outlook.com , k.necludov@gmail.com .

*Accepted at the 1st Machine Learning for Life and Material Sciences Workshop at ICML 2024.* Copyright 2024 by the author(s).

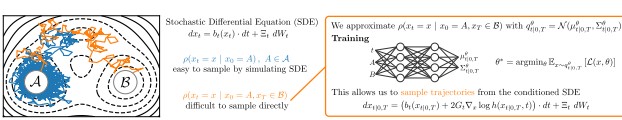

Figure 1: Given reference dynamics, transition path sampling seeks to capture the conditional or posterior distribution over paths which reach a terminal set $x_T \in \mathcal{B}$. However, simulating the reference dynamics (blue) can be wasteful since we rarely obtain paths (orange) which reach (the vicinity) of the terminal set $\mathcal{B}$. This is a major challenge for techniques based on importance sampling or Monte Carlo estimation, even when adding a control term to the reference dynamics. By contrast, our approach optimizes a tractable variational distribution over transition paths with a parameterization satisfying initial and terminal conditions by design.

cations from generative modeling (De Bortoli et al., 2021; Chen et al., 2021a; Liu et al., 2022; 2023c; Somnath et al., 2023) to molecular simulation (Anderson, 2007; Wu et al., 2022; Plainer et al., 2023; Holdijk et al., 2024), drug discovery (Kirmizialtin et al., 2012; 2015; Dickson, 2018), and materials science (Xi et al., 2013; Selli et al., 2016).

**Transition Path Sampling.** In this work, we take a particular interest in the problem of *transition path sampling* (TPS) in computational chemistry (Dellago et al., 2002; Weinan and Vanden-Eijnden, 2010), which attempts to describe how molecules transition between local energy minima or metastable states under random fluctuations or the influence of external forces. Understanding such transitions has numerous applications for combustion, catalysis, battery, material design, and protein folding (Zeng et al., 2020; Klucznik et al., 2024; Blau et al., 2021; Noé et al., 2009; Escobedo et al., 2009). While the TPS problem is often framed as finding the 'most probable path' transitioning between states (Dürr and Bach, 1978; Vanden-Eijnden and Heymann, 2008), we draw explicit connections to the Doob's $h$-transform and seek to match the *full* posterior distribution over conditioned processes.

**Doob's $h$-Transform.** For Brownian motion diffusion processes, conditioning is known to be achieved by Doob's $h$-transform (Doob, 1957; Särkkä and Solin, 2019). However, solving this problem amounts to estimating rare event probabilities or matching a complex target distribution. Approaches which involve simulation of trajectories to construct Monte Carlo expectations or importance sampling

estimators (Papaspiliopoulos and Roberts, 2012; Schauer et al., 2017; Holdijk et al., 2024) can be extremely inefficient if the target event is rare or endpoint distribution is difficult to match. Recent methods based on score matching (Heng et al., 2021) or nonlinear Feynman-Kac (Chopin et al., 2023) still require simulation within an inner optimization loop.

**Variational Formulation of Doob's $h$-Transform.** In this work, we propose a variational formulation of Doob's $h$-transform as the solution to an optimization on the space of paths of probability distributions (Thm. 1). We focus on solving for the Doob transform conditioning on a particular terminal point, which is natural in the TPS setting (see Fig. 1). Taking inspiration from recent bridge matching methods (Peluchetti, 2021; 2023; Liu et al., 2022; Lipman et al., 2022; Shi et al., 2023; Liu et al., 2023a), we propose a parameterization with the following attractive features.

1. **Every Sample Matters.** In contrast to most existing approaches, our method is *simulation-free*, thereby avoiding computationally wasteful simulation methods to estimate rare-event probabilities and inefficient importance or rejection sampling. We thus refer to our approach as being *sample-efficient*.

2. **Optimization over Sampling.** To approximate the conditioned process, we propose an expressive variational family which is tractable to sample and can be optimized end-to-end using neural networks.

3. **Problem-Informed Parameterization.** Our parameterization enforces the boundary conditions *by design*, thereby reducing the search space for optimization and efficiently making use of the conditioning information.

## 2. Background

### 2.1. Transition Path Sampling
Consider a forward or reference stochastic process with states $x_t$ and transition probability $\rho(x_{t+dt} = y \mid x_t = x)$. Starting from an initial $x_0 = A$, the density of the path is

$$\rho(x_T, \ldots, x_{dt} \mid x_0 = A) = \prod_{t=dt}^{T-dt} \rho(x_{t+dt} \mid x_t) \cdot \rho(x_{dt} \mid x_0 = A)$$

The problem of rare event sampling aims to condition this reference stochastic process on some event at time $T$, for example, that the final state belongs to a particular set $x_T \in \mathcal{B}$. We are interested in sampling from the entire *transition path*, i.e. the posterior distribution over intermediate states

$$\rho(x_{T-dt}, \ldots, x_{dt} \mid x_0 = A, x_T \in \mathcal{B}) = \frac{\rho(x_T \in \mathcal{B}, x_{T-dt} \ldots, x_{dt} \mid x_0 = A)}{\rho(x_T \in \mathcal{B} \mid x_0 = A)}.$$

Moving to continuous time, we focus on the transition path sampling problem in the case where the reference process is given by a Brownian motion. In particular, we are motivated by applications in computational chemistry (Dellago

et al., 2002; Weinan and Vanden-Eijnden, 2010), where the reference process is given by molecular dynamics following either overdamped Langevin dynamics,

$$dx_t = -(\gamma M)^{-1}\nabla_x U(x_t) \cdot dt + (\gamma M)^{-1/2}\sqrt{2k_B \mathcal{T}} \cdot dW_t, \quad (1)$$

or the second-order Langevin dynamics with spatial coordinates $\bar{x}_t$ and velocities $\bar{v}_t$,

$$d\bar{x}_t = \bar{v}_t \cdot dt, \quad (2)$$
$$d\bar{v}_t = \left(-M^{-1}\nabla_x U(\bar{x}_t) - \gamma M^{-1}\bar{v}_t\right) \cdot dt + M^{-1/2}\sqrt{2\gamma k_B \mathcal{T}} \cdot dW_t.$$

where $W_t$ denotes the Wiener process. Note, $U$ is a potential energy, $k_B \mathcal{T}$ is the Boltzman constant times temperature, $M$ is the mass matrix, and $\gamma$ is the friction coefficient.

### 2.2. Doob's $h$-transform
Doob's $h$-transform addresses the question of conditioning a reference Brownian motion to satisfy a terminal condition such as $x_T \in \mathcal{B}$, thereby providing an avenue to solve the transition path sampling problem described above. Without loss of generality, and to provide a unified treatment of the dynamics in (1)–(2), we consider the forward or reference stochastic differential equation (SDE),

$$\mathbb{P}_{0:T}^{\text{ref}}: \quad dx_t = b_t(x_t) \cdot dt + \Xi_t\, dW_t, \quad x_0 \sim \rho_0(x), \quad (3)$$

with drift $b_t : \mathbb{R}^N \to \mathbb{R}^N$ and diffusion matrix $\Xi_t \in \mathbb{R}^{N \times N}$ such that $G_t := \frac{1}{2}\Xi_t\Xi_t^T$ is positive definite.[1] We denote the induced path measure as $\mathbb{P}_{0:T}^{\text{ref}} \in \mathcal{P}(\mathcal{C}([0, T] \to \mathbb{R}^N))$.

Remarkably, Doob's $h$-transform (Doob, 1957; Särkkä and Solin, 2019, Sec. 7.5) shows that conditioning the reference process (3) on $x_T \in \mathcal{B}$ yields another Brownian motion.

**Proposition 1.** [Jamison (1975, Thm. 2)] *Let* $h(x,t) := \rho(x_T \in \mathcal{B} \mid x_t = x)$ *denote the conditional transition density with respect to the reference process in* (3). *Letting* $G_t := \frac{1}{2}\Xi_t\Xi_t^T$, *the SDE*

$$dx_{t|0,T} = \left(b_t(x_{t|0,T}) + 2G_t\nabla_x \log h(x_{t|0,T}, t)\right) \cdot dt + \Xi_t\, dW_t \quad (4)$$

*is associated with the following transition probabilities*

$$\rho(x_t = y \mid x_s = x, x_T \in \mathcal{B}) = \frac{h(y, s)}{h(x, t)}\rho(x_t = y \mid x_s = x), \quad (5)$$

*for* $s < t < T$, *where we omit the dependence of* $h(x, t)$ *on* $\mathcal{B}$ *for simplicity of notation.*

See App. C.1 for proof. The conditioned transition probabilities in (5) allow us to directly construct the transition path in Sec. 2.1. Using Bayes rule, we have

$$\frac{\rho(x_{T-dt}, \ldots, x_{dt} \mid x_0 = A, x_T \in \mathcal{B})}{\rho(x_{T-dt} \ldots, x_{dt} \mid x_0 = A)} = \frac{h(x_{T-dt}, T - dt)}{h(A, 0)}$$

Thus, we can solve the TPS problem by exactly solving for the $h$-function and simulating the SDE in (4).

---

[1] See (19) in App. B.1 to write (2) in the form of (3).

**Theorem 1.** *The following Lagrangian action minimization has a unique solution which matches Doob's h-transform in* *Prop. 1, where the optimal* $q_{t|0,T}^*(x)$ *and* $v_{t|0,T}^*(x) = \nabla_x \log h(x,t)$ *satisfy the PDEs in* *App. C.1 Prop. 3,*

$$\mathcal{S} = \min_{q,v} \int_0^T dt \int dx \, q_{t|0,T}(x)\langle v_{t|0,T}(x), G_t \, v_{t|0,T}(x)\rangle, \tag{6a}$$

$$s.t. \quad \frac{\partial q_{t|0,T}(x)}{\partial t} = -\langle \nabla_x, q_{t|0,T}(x)\big(b_t(x) + 2G_t \, v_{t|0,T}(x)\big)\rangle + \sum_{ij}(G_t)_{ij}\frac{\partial^2}{\partial x_i \partial x_j}q_{t|0,T}(x), \tag{6b}$$

$$q_0(x) = \delta(x - A), \qquad q_T(x) = \delta(x - B). \tag{6c}$$

Finally, the $h$-process and marginal density of the conditioned process satisfy a set of forward and backward Kolmogorov equations. These are crucial for deriving our variational objective, but deferred to App. A.1 Prop. 3 for space.

## 3. Method

### 3.1. Doob's Lagrangian

Consider reference dynamics in the form of either (1) or (2), with known drift $b_t$ or energy $U$. We restrict our attention to conditioning on a terminal rare event of reaching a given endpoint $x_T = B$, along with an initial point $x_0 = A$. We approach finding Doob's $h$-transform via a *least action principle* where, in the Thm. 1, we define a Lagrangian action whose minimization yields the optimal $q_{t|0,T}^*(x) = \rho_{t|0,T}(x)$ and $v_{t|0,T}^*(x) = \nabla_x \log h(x,t)$ from Prop. 1.

This objective will form the basis for our computational approach, with proof of Thm. 1 deferred to App. C.2. We provide additional analysis of our objective in App. C.1.

**Challenges of Optimizing (6a).** We highlight several distinctive features of our problem which inform the development of computational methods in Sec. 3.2.

1. First, (6a) requires that we are able to efficiently sample from the conditioned process in (13) or $q_{t|0,T}$. This appears challenging due to the nonlinearity of both the reference and variational drifts, $b_t$ and $v_{t|0,T}$.

2. For a given $q_{t|0,T}$, it can be difficult to solve for $v_{t|0,T}$ which satisfies the Fokker-Planck equation in (6b).

3. Finally, we would like to strictly enforce the boundary constraints on $q_{t|0,T}$ or $\mathbb{Q}_{0:T}^v$ to avoid simulating or wasting computation on trajectories for which $x_T \neq B$.

Our parameterization of $q_{t|0,T}$ will *avoid simulation* of the SDE (13) (Challenge 1), provide *analytic solutions* for $v_{t|0,T}$ satisfying (6b) with given $q_{t|0,T}$ (Challenge 2), and enforce the boundary constraints *by design* (Challenge 3).

### 3.2. Computational Approach

We now propose a family of Gaussian (mixture) path parameterizations $q_{t|0,T}$ which overcome the aforementioned

computational challenges, while still maintaining expressivity. We present all aspects of our proposed method in the context of the first-order dynamics and defer extensions to mixture paths and second-order dynamics to App. B.

**Tractable Drift** $v_{t|0,T}$ **for Variational Doob Objective.** Consider modifying the Fokker-Planck constraint in (6b), to absorb all drift terms into a single vector field $u_{t|0,T}$,

$$\frac{\partial q_{t|0,T}(x)}{\partial t} = -\langle \nabla_x, q_{t|0,T}(x) u_{t|0,T}(x)\rangle + \sum_{ij}(G_t)_{ij}\frac{\partial^2}{\partial x_i \partial x_j}q_{t|0,T}(x). \tag{7}$$

To address Challenge 2, we restrict attention to variational families of $q_{t|0,T} \in \mathcal{Q}$ where it is *analytically tractable* to calculate a vector field $u_{t|0,T}^{(q)}$ which satisfies (7). We first consider the family of Gaussian paths $\mathcal{Q}_G$, in similar fashion to flow matching methods (Lipman et al., 2022; Tong et al., 2023; Liu et al., 2023a), with proof in App. B.

**Proposition 2.** *For the family of endpoint-conditioned marginals* $q_{t|0,T}(x) = \mathcal{N}(x \,|\, \mu_{t|0,T}, \Sigma_{t|0,T})$,

$$u_{t|0,T}^{(q)}(x) \coloneqq \frac{\partial \mu}{\partial t} + \left[\frac{1}{2}\frac{\partial \Sigma}{\partial t}\Sigma_{t|0,T}^{-1} - G_t \Sigma_{t|0,T}^{-1}\right](x - \mu_{t|0,T}) \tag{8}$$

*satisfies the Fokker-Planck equation* (7) *for* $q_{t|0,T}$ *and diffusion coefficients* $G_t = \frac{1}{2}\Xi_t\Xi_t^T$.

Given $u_{t|0,T}^{(q)}$ corresponding to $q_{t|0,T}$, we can simply solve for the $v_{t|0,T}$ satisfying the Fokker-Planck euqation in (6b) in our variational Doob objective (Thm. 1). Since $G_t$ was assumed to be invertible and the base drift $b_t$ is known,

$$v_{t|0,T}^{(q)}(x) = \frac{1}{2}(G_t)^{-1}\left(u_{t|0,T}^{(q)}(x) - b_t(x)\right), \tag{9}$$

We may now evaluate terms involving $v_{t|0,T}$ in our Lagrangian objective in (6) using (9) directly, without solving an inner minimization over $v_{t|0,T}$ (addressing Challenge 2).

**Optimization over** $q_{t|0,T}$ **satisfying (6c).** Given the ability to evaluate $v_{t|0,T}^{(q)}$ for a given $q_{t|0,T} \in \mathcal{Q}_G$ as above, our variational Doob objective in (6a) reduces to a single optimization over the marginals $q_{t|0,T}$ of a conditioned process which satisfies the boundary conditions (6c).

We consider parameterizing the mean $\mu_{t|0,T}$ and covariance $\Sigma_{t|0,T}$ of our Gaussian path $q_{t|0,T}$ using a neural network.

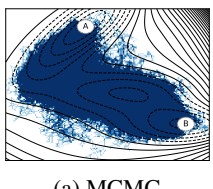 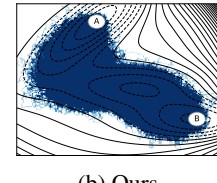 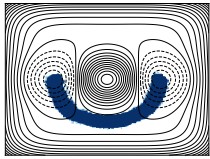 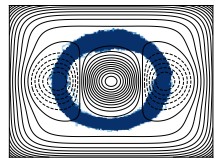

| (a) MCMC | (b) Ours |
|---|---|

Figure 2: Comparing path histograms of TPS using fixed-length two-way shooting and comparing it with our variational approach.

| (a) Single Gaussian | (b) Mixture of Gaussians |
|---|---|

Figure 3: Expressivity of Gaussian vs. mixture of Gaussian paths on a symmetric potential with two transition path modes.

For simplicity, we consider a diagonal parameterization $\Sigma_{t|0,T} = \texttt{diag}(\{\sigma^2_{t|0,T,d}\}^D_{d=1})$. We parameterize a neural network $\text{NNET}_\theta : [0,T] \times \mathbb{R}^D \times \mathbb{R}^D \to \mathbb{R}^D \times \mathbb{R}^D$ which inputs $(t, A, B)$. Its output is used to construct

$$x_{t|0,T} = \mu^{(\theta)}_{t|0,T} + \Sigma^{(\theta)}_{t|0,T}\, \epsilon, \text{ where } \epsilon \sim \mathcal{N}(0, \mathbb{I}_D). \quad (10)$$

$$\mu^{(\theta)}_{t|0,T} = (1-t)A + t\,B + t(1-t)\text{NNET}_\theta(t,A,B)_{[:D]}$$

$$\Sigma^{(\theta)}_{t|0,T} = t(1-t)\texttt{diag}\big(\text{NNET}_\theta(t,A,B)_{[D:]}\big) + \sigma^2_{\min}\mathbb{I}.$$

Crucially, our Gaussian parameterization addresses Challenge 1, in that we can easily draw samples $x_{t|0,T} \sim q_{t|0,T}$ from our variational conditioned process (6b) *without simulating* the corresponding SDE with nonlinear drift (13). Further, $t(1-t)$ coefficients in (10) ensure that the (smoothed) boundary conditions are satisfied by design (Challenge 3). We add $\sigma^2_{\min}$ to ensure invertibilty of $\Sigma_{t|0,T}$ (see (8)) as $t \to 0$ or $t \to T$, but preserve $q_0(x_0) = \mathcal{N}(x_0 \mid A, \sigma^2_{\min}\mathbb{I}_D) \approx \delta(x_0 - A)$ and $q_T(x_T) = \mathcal{N}(x_T \mid B, \sigma^2_{\min}\mathbb{I}_D) \approx \delta(x_T - B)$.

**Reparameterization Gradients.** Since we have now shown that our parameterization satisfies the constraints (6b)-(6c) by design, we can finally optimize our variational Doob objective with respect to $q_{t|0,T} \in \mathcal{Q}_G$ using the reparameterization trick (Kingma and Welling, 2013; Rezende et al., 2014). In particular, for the expectation at each $t$ in (6a),

$$\nabla_\theta \mathbb{E}_{q(x)}\Big[\big\langle v^{(q,\theta)}_{t|0,T}(x), G_t\, v^{(q,\theta)}_{t|0,T}(x)\big\rangle\Big] \quad (11)$$

$$= \mathbb{E}_{\mathcal{N}(\epsilon|0,\mathbb{I})}\Big[\nabla_\theta\big\langle v^{(q,\theta)}_{t|0,T}\big((g(t,\epsilon;\theta)\big), G_t\, v^{(q,\theta)}_{t|0,T}\big((g(t,\epsilon;\theta)\big)\big\rangle\Big]$$

where $x = g(t, \epsilon; \theta)$ is the parameterized map in (10) and $v^{(q,\theta)}_{t|0,T}$ depends on $\theta$ via $\mu^{(\theta)}_{t|0,T}$, $\Sigma^{(\theta)}_{t|0,T}$ in (8)–(9). In practice, we approximate gradients using a single sample of $\epsilon$ at uniformly sampled discrete time points $0 \le t \le T$ which represent physical time (e.g., femtoseconds).

# 4. Experiments

We investigate the capabilities of our approach across a variety of different settings. We first illustrate features of our method on toy examples before continuing to real-world molecular systems, including a commonly-used benchmark system, alanine dipeptide, and a small protein, Chignolin. The code behind our method is available at the following link. Before diving into the experiments, we introduce the evaluation procedure and baseline methods.

**Evaluation metrics.** In our evaluation, we emphasize two key quantities: accuracy and efficiency. Efficiency is evaluated by the number of calls to the potential energy function, which requires extensive computation and dominates the runtime of larger molecules. For accuracy, we evaluate the log-likelihood of each sampled path and the maximum energy point (saddle point/transition state) along each sampled path. A good method samples many probable paths (i.e., high log-likelihood) and an accurate transition state (i.e., small maximum energy). See App. E for further details.

**Baselines.** We compare our approach against the Markov Chain Monte Carlo (MCMC)-based two-way shooting method with uniform point selection with variable or fixed length trajectories. We found that two-way shooting produced the most diverse path ensembles among possible baselines, although the acceptance probability can be relatively low for systems dominated by diffusive dynamics (Brotzakis and Bolhuis, 2016) and might be improved by learning shooting point selection. This baseline gives theoretical guarantees about the ensemble and thus can be considered as a proxy for the ground truth.

## 4.1. Synthetic Datasets

**Müller-Brown Potential.** The Müller-Brown potential is a popular benchmark to study transition path sampling between metastable states. It consists of three local minima, and we aim to sample transition paths connecting state $A$ and state $B$ with a circular state definition. In Fig. 2, we visualize the potential and the sampled paths and can see that the same ensemble is sampled for both our method and two-way shooting. Our method exhibits a slightly reduced variance for unlikely transitions. In Table 1, we can observe that MCMC-based methods require many potential evaluations to achieve a good result, which comes from the low acceptance rate (especially when fixing the lengths of trajectories). Our method requires fewer energy evaluations (1 million vs. 1 billion) while finding paths with similar energy and likelihood. Note, the likelihood for variable approaches has been omitted, as it is governed by the number of steps in the trajectory and cannot be compared directly.

**Gaussian Mixture.** We further consider a potential in which the states are separated by a symmetric high-energy barrier

| Method | # Evalss (↓) | Max Energy (↓) | MinMax Energy (↓) | Log-Lkd (↑) | Max Log-Lkd (↑) |
|---|---|---|---|---|---|
| MCMC (variable) | 3.53M | -13.77 ± 16.43 | -40.75 | - | - |
| MCMC | 1.03B | -17.80 ± 14.77 | -40.21 | 866.56 ± 17.00 | 907.15 |
| Ours | 1.28M | -14.81 ± 13.73 | -40.56 | 858.50 ± 17.61 | 909.74 |

Table 1: Transition path sampling experiment for Müller-Brown potential. We report the number of potential evaluations needed to sample 1,000 paths, as well as the maximum energy and the likelihood of each path (including mean and standard deviation). MinMax energy reports the lowest maximum energy of all paths.

| Method | States | # Evaluations (↓) | Max Energy (↓) | MinMax Energy (↓) |
|---|---|---|---|---|
| MCMC (variable length) | CV | 25.82M | 1,212.81 ± 19,444.46 | 28.67 |
| MCMC* | CV | 1.29B | 288.46 ± 128.31 | 60.52 |
| MCMC (variable length) | relaxed | 80.23M | 269.16 ± 248.51 | 39.11 |
| MCMC | relaxed | N/A | N/A | N/A |
| MCMC (variable length) | exact | N/A | N/A | N/A |
| MCMC | exact | N/A | N/A | N/A |
| Ours (Cartesian) | exact | 38.40M | 804.24 ± 0.20 | 803.62 |
| Ours (Cartesian, 2 Mixtures) | exact | 51.20M | 828.77 ± 27.34 | 803.44 |
| Ours (Internal) | exact | 51.20M | 352.20 ± 0.04 | 352.08 |
| Ours (Internal, 2 Mixtures) | exact | 51.20M | 371.16 ± 82.88 | 239.66 |

Table 2: Transition path sampling for alanine dipeptide. For MCMC methods, we compare different state definitions of $\mathcal{A}, \mathcal{B}$: 'CV' uses $\phi, \psi$ angles. 'Exact' uses a very small threshold of aligned root-mean-square deviation (RMSD) around reference states $A, B$ (as in Ours). 'Relaxed' uses a larger threshold of RMSD around $A, B$. The method marked with a * only samples 100 paths due to computational limitations, while others sample 1,000. Fields with N/A are intractable as they require significantly more than 1 billion potential evaluations.

that allows for two distinct reaction channels. In Fig. 3, we observe that a single Gaussian path cannot model a system with multiple modes of transition paths. Nevertheless, this issue can be resolved using a mixture of Gaussian paths, with slightly increased computational cost.

## 4.2. Second-order Dynamics and Molecular Systems

**Experiment Setup.** We evaluate our methods on real-world high-dimensional molecular systems governed by the second-order dynamics (2): *alanine dipeptide* and *Chignolin*. Alanine dipeptide is a well-studied system of 22 atoms (66 total degrees of freedom), where the molecule can be described by two collective variables (CV): the dihedral angles $\phi, \psi$. Chignolin is a larger system consisting of 10 residues with 138 atoms (414 total degrees of freedom) that cannot be summarized as easily. We use an AMBER14 forcefield (Maier et al., 2015) implemented in OpenMM (Eastman et al., 2017) but use DMFF (Wang et al., 2023) to backpropagate through the energy evaluations.

**Alanine Dipeptide.** In Table 2, we report results for four variants of our models, which either predict Cartesian coordinates or internal coordinates in the form of bond lengths and dihedral angles (compare App. E), either with or without Gaussian mixture. For our method, operating in internal coordinates takes more iterations to converge but generates better results compared to Cartesian coordinates, where internal coordinates have nicely distributed input and our network does need not learn equivariances (Du et al., 2022). Similarly, Gaussian mixture paths perform slightly better

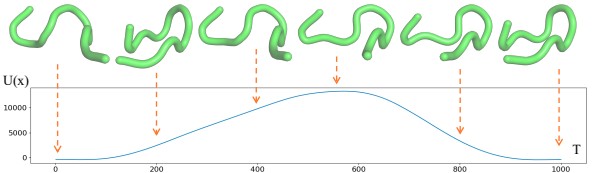

Figure 4: Transition path for the protein Chignolin. The energy plot demonstrates that the conformation goes through a high energy barrier in a total of $T = 1,000\,fs$, with the highest energy state reached at $567\,fs$.

than a single Gaussian path due to the extra expressiveness. We note that paths sampled with Gaussian mixture exhibit a larger variance in max energy as they represent multiple reaction channels.

We find that prior-informed definitions of the desired initial and target states (i.e., CV) are necessary for MCMC to work efficiently with fixed-length trajectories. Finding these CVs in practice is challenging and only possible in this instance because the molecule is small and well-studied. For the larger system size in Table 2, it becomes intractable to use MCMC for reaching precise states $A, B$ ('exact') instead of larger regions ('relaxed'), or for computing fixed-length trajectories. Variable length MCMC with relaxed endpoint conditions and fixed-length MCMC with CV perform well on this task, but our method is competitive using fewer evaluations and more strict boundary conditions.

**Chignolin.** The folding dynamics of Chignolin already pose a challenge and have not yet been well-studied compared to alanine dipeptide. We illustrate the qualitative experimental results for this system in Fig. 4. Operating in Cartesian space, our model samples a feasible transition within 12.8M potential energy evaluation calls and a transition with a duration of $T = 1\,ps$, which is faster compared to $T = 0.6\,\mu s$ in Lindorff-Larsen et al. (2011).

## 5. Conclusion, Limitations and Future Work

In this paper, we propose an efficient computational framework for transition path sampling with Brownian dynamics. We formulate the transition path sampling problem by using Doob's $h$-transform to condition a reference stochastic process, and propose a variational formulation for efficient optimization. Specifically, we propose a simulation-free training objective and model parameterization that imposes boundary conditions as hard constraints. We compare our methods with MCMC-based baselines and show comparable accuracy with lower computational costs on both synthetic datasets and real-world molecular systems. Finally, our method might be improved or extended by (1) accounting for conditioning on a set of terminal events, (2) amortizing over many state pairs or systems and finally learning an unconditioned process, and (3) accommodating variable length paths.

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

# A. Doob's $h$-Transform and its Variational Formulation

## A.1. Forward-Backward Kolmogorov Equations for Doob's $h$-Transform

**Proposition 3.** *The following PDEs are obeyed by (a) the marginal density of the conditioned process $\rho_{t|0,T}(x) = \rho(x_t = x \mid x_0 = A, x_T \in \mathcal{B})$ and (b) the $h$-function $h(x,t)$ (which implicitly depends on $\mathcal{B}$),*

$$\frac{\partial \rho_{t|0,T}(x)}{\partial t} + \left\langle \nabla_x, \rho_{t|0,T}(x)\big(b_t(x) + 2G_t\nabla_x \log h(x,t)\big)\right\rangle - \sum_{ij}(G_t)_{ij}\frac{\partial^2}{\partial x_i \partial x_j}\rho_{t|0,T}(x) = 0\,, \tag{12a}$$

$$\frac{\partial h(x,t)}{\partial t} + \left\langle \nabla_x h(x,t), b_t(x)\right\rangle + \sum_{ij}(G_t)_{ij}\frac{\partial^2}{\partial x_i \partial x_j}h(x,t) = 0\,. \tag{12b}$$

*Reparameterizing* (12b) *in terms of $s(x,t) := \log h(x,t)$, we can also write*

$$\frac{\partial s(x,t)}{\partial t} + \left\langle \nabla s(x,t), G_t\nabla s(x,t)\right\rangle + \left\langle \nabla s(x,t), b_t(x_t)\right\rangle + \sum_{ij}(G_t)_{ij}\frac{\partial^2}{\partial x_i \partial x_j}s(x,t) = 0. \tag{12c}$$

We provide proof of this proposition in App. C.1. We recall the simple example of the Brownian bridge, although our methods will handle the case of nonlinear reference processes as in (1)-(2).

**Example 1** (Brownian Bridge). *Consider a reference process $dx_t = dW_t$. Conditioning on $x_T = B$ for a particular value of $B$, the Doob $h$-transform amounts to a Gaussian transition kernel $h(x,t) := \rho(x_T = B \mid x_t = x) = \mathcal{N}(x_T \mid x_t, T - t)$. Plugging the log gradient $\nabla_x \log h(x,t)$ into (4) yields the conditioned SDE known as the Brownian bridge, $dx_{t|0,T} = (x_T - x_{t|0,T})/(T - t)dt + dW_t$.*

## A.2. Analysis of the Variational Objective in Thm. 1

**Unconstrained Dual Objective.** Introducing Lagrange multipliers to enforce the constraints in (6b)–(6c) and eliminating $v_{t|0,T}$, we obtain an alternative, unconstrained version of (6a).

**Corollary 1.** *The Lagrangian objective in Thm. 1 which solves Doob's $h$-transform is equivalent to*

$$\mathcal{S} = \min_{q_{t|0,T}} \max_{s} \; s(B,1) - s(A,0) - \int_0^1 dt \int dx \, q_{t|0,T}\left(\frac{\partial s}{\partial t} + \langle\nabla s, G_t\nabla s\rangle + \langle\nabla s, b_t\rangle + \langle\nabla, G_t\nabla s\rangle\right)$$

*if $q_{t|0,T}$ satisfies* (6c). *Note $v_{t|0,T}(x) = \nabla_x s(x,t)$, with $s^*(x,t) = \log h(x,t)$ at optimality.*[2]

This objective is similar to the objectives optimized by Action Matching methods (Neklyudov et al., 2023; 2024). Notably, the objective in Cor. 1 is expressed *directly* in terms of the (log) of the $h$-function for fixed conditioning information $x_T = B$. We also note that the Hamilton Jacobi-style quantity, whose expectation appears in the final term, is zero at optimality in (12c) of Prop. 3.

**Path Measure Perspective.** We next interpret our variational objective in Thm. 1 as minimizing a KL divergence over path measures. Let $\mathbb{P}_{0:T}^{\text{ref}}$ denote the law of the reference SDE in (3) with fixed $\mathbb{P}_0^{\text{ref}} = \delta(x_0 - A)$. Let $\mathbb{Q}_{0:T}^v$ denote the law of a controlled process similar to (4), but with a variational $v_{t|0,T}$ in place of $\nabla_x \log h$,

$$\mathbb{Q}_{0:T}^v: \qquad dx_t = \big(b_t(x_t) + 2G_t\, v_{t|0,T}(x_{t|0,T})\big) \cdot dt + \Xi_t\, dW_t\,, \qquad x_0 = A. \tag{13}$$

Note that the density $q_{t|0,T}$ of $\mathbb{Q}_{0:T}^v$ satisfies the Fokker-Planck equation in (6b) (Särkkä and Solin, 2019, Sec. 5.2) Using the Girsanov Theorem, the objective in (6a) can then be viewed as a KL divergence minimization over path measures $\mathbb{Q}_{0:T}^v$ which satisfy the boundary constraints.

**Corollary 2.** *The Lagrangian objective in Thm. 1 is equivalent to the following optimization of $\mathbb{Q}_{0:T}^v$*

$$\mathcal{S} := \min_{\mathbb{Q}_{0:T}^v \; s.t. \; \mathbb{Q}_0^v = \delta_A, \mathbb{Q}_T^v = \delta_B} D_{KL}[\mathbb{Q}_{0:T}^v : \mathbb{P}_{0:T}^{ref}] \tag{14}$$

*where the minimizing argument recovers the path measure $\mathbb{P}_{0:T}^*$ associated with the SDE in* (4).

---

[2]Again, note that we omit the dependence of $s(x,t)$ and $h(x,t)$ on the conditioning information $B$.

Our Lagrangian action minimization thus solves a Schrödinger Bridge (SB) problem (Schrödinger, 1932; Léonard, 2014) with Dirac delta functions as the endpoint measures. Our objective in (6a) particularly resembles optimal control formulations of SB (Chen et al., 2016; 2021b). While it is well-known that the Doob $h$-transforms (and large deviations more generally) play a role in the solution to SB problems (e.g. Jamison (1975); Léonard (2014)), our interest in the transition path sampling problem leads to specific computational decisions below. See App. D for further discussion.

## B. Gaussian Path Parameterizations

We begin by proving Prop. 2 in the main text, before discussing how our algorithm extends to second order dynamics (App. B.1) and mixtures of Gaussians (App. B.2).

**Proposition. 3.** *For the family of endpoint-conditioned marginals $q_{t|0,T}(x) = \mathcal{N}(x \,|\, \mu_{t|0,T}, \Sigma_{t|0,T})$,*

$$u_{t|0,T}^{(q)}(x) := \frac{\partial \mu_{t|0,T}}{\partial t} + \left[ \frac{1}{2} \frac{\partial \Sigma_{t|0,T}}{\partial t} \Sigma_{t|0,T}^{-1} - G_t \, \Sigma_{t|0,T}^{-1} \right] \left( x - \mu_{t|0,T} \right) \tag{15}$$

*satisfies the Fokker-Planck equation (7) for $q_{t|0,T}$ and diffusion coefficients $G_t = \frac{1}{2} \Xi_t \Xi_t^T$.*

*Proof.* Consider the following identities for the Gaussian family of marginals $q_t(x) = \mathcal{N}(x|\mu_t, \Sigma_t)$, where we omit conditioning $q_t \leftarrow q_{t|0,T}$ for simplicity of notation,

$$\log q_t(x) = -\frac{1}{2}(x - \mu_t)^T \Sigma_t^{-1}(x - \mu_t) - \frac{d}{2} \log(2\pi) - \frac{1}{2} \log \det \Sigma_t \,, \tag{16a}$$

$$\nabla_x \log q_t(x) = -\Sigma_t^{-1}(x - \mu_t) \,, \tag{16b}$$

$$\frac{\partial}{\partial t} \log q_t(x) = (x - \mu_t)^T \Sigma_t^{-1} \frac{\partial \mu_t}{\partial t} + \frac{1}{2}(x - \mu_t)^T \Sigma_t^{-1} \frac{\partial \Sigma_t}{\partial t} \Sigma_t^{-1}(x - \mu_t) - \frac{1}{2} \mathrm{tr}\left( \Sigma_t^{-1} \frac{\partial \Sigma_t}{\partial t} \right) \tag{16c}$$

We begin by solving for a vector field $u_t^{\mathrm{o}}(x)$ that satisfies the continuity equation (where $u_t^{\mathrm{o}}$ denotes the drift of an ODE)

$$\frac{\partial q_t}{\partial t} = -\langle \nabla_x, q_t u_t^{\mathrm{o}} \rangle = -q_t \langle \nabla_x, u_t^{\mathrm{o}} \rangle + \langle \nabla_x q_t, \nabla_x u_t^{\mathrm{o}} \rangle$$

$$\implies \frac{\partial}{\partial t} \log q_t = -\langle \nabla_x, u_t^{\mathrm{o}} \rangle - \langle \nabla_x \log q_t, u_t^{\mathrm{o}} \rangle \tag{17}$$

The vector field satisfying this equation is

$$u_t^{\mathrm{o}}(x) = \frac{\partial \mu_t}{\partial t} + \frac{1}{2} \frac{\partial \Sigma_t}{\partial t} \Sigma_t^{-1}(x - \mu_t) \tag{18}$$

which we can confirm using the identities in (16). Indeed, for the terms on the RHS of Eq. (17),

$$-\langle \nabla_x, u_t^{\mathrm{o}} \rangle = -\frac{1}{2} \mathrm{tr}\left( \Sigma_t^{-1} \frac{\partial \Sigma_t}{\partial t} \right) \,,$$

$$-\langle \nabla_x \log q_t, u_t^{\mathrm{o}} \rangle = \left\langle \Sigma_t^{-1}(x - \mu_t), \frac{\partial \mu_t}{\partial t} \right\rangle + \frac{1}{2}(x - \mu_t)^T \Sigma_t^{-1} \frac{\partial \Sigma_t}{\partial t} \Sigma_t^{-1}(x - \mu_t) \,.$$

Putting these terms and the time derivative from (16c) into Eq. (17) we conclude the proof.

However, we are eventually interested in finding the formula for the drift $u_t$ that satisfies the Fokker-Planck equation in (7). That is, to describe the same evolution of density $\frac{\partial q_t(x)}{\partial t}$, the relationship between $u_t$ and $u_t^{\mathrm{o}}$ is as follows

$$\frac{\partial q_t(x)}{\partial t} = -\langle \nabla_x, q_t u_t^{\mathrm{o}} \rangle = -\langle \nabla_x, q_t \, u_t \rangle + \langle \nabla_x, G_t \nabla_x q_t \rangle$$

$$= -\langle \nabla_x, q_t \, u_t \rangle + \langle \nabla_x, G_t q_t \nabla_x \log q_t \rangle$$

$$= -\left\langle \nabla_x, q_t \underbrace{(u_t - G_t \nabla_x \log q_t)}_{u_t^{\mathrm{o}}} \right\rangle$$

Finally, we use the identities in (16) to obtain

$$u_t = u_t^o + G_t \nabla_x \log q_t = \frac{\partial \mu_t}{\partial t} + \frac{1}{2} \frac{\partial \Sigma_t}{\partial t} \Sigma_t^{-1}(x - \mu_t) - G_t \Sigma_t^{-1}(x - \mu_t)$$

$$\implies \quad u_t = \frac{\partial \mu_t}{\partial t} + \left[ \frac{1}{2} \frac{\partial \Sigma_t}{\partial t} \Sigma_t^{-1} - G_t \Sigma_t^{-1} \right] (x - \mu_t)$$

$\square$

### B.1. Second-Order Dynamics

To handle the case of the second-order dynamics in (2), we can adapt our recipe from the previous section with minimal modifications by extending the state space $x \in \mathbb{R}^D$ to include velocities $\bar{v}$, with $x = (\bar{x}, \bar{v}) \in \mathbb{R}^{2D}$. However, note that the dynamics in (2) are no longer stochastic in the spatial coordinates $\bar{x}$. To ensure invertibility of $G_t$ and existence of the $h$-transform, we add a small nonzero diffusion coefficient in the coordinate space $\bar{x}$, so that the reference process in Eq. (3) is given by

$$x_t = \begin{bmatrix} \bar{x}_t \\ \bar{v}_t \end{bmatrix}, \quad b_t(x_t) = \begin{bmatrix} \bar{v}_t \\ -M^{-1}\nabla_x U(\bar{x}_t) - \gamma M^{-1}\bar{v}_t \end{bmatrix}, \quad \Xi_t = \begin{bmatrix} \xi_{\min}\mathbb{I}_D & 0 \\ 0 & M^{-1/2}\sqrt{2\gamma k_B \mathcal{T}} \end{bmatrix}. \tag{19}$$

All steps in our algorithm proceed in similar fashion to Sec. 3.2. We now parameterize $q_{t|0,T}(\bar{x}, \bar{v})$ using NNET$_\theta$ : $[0, T] \times \mathbb{R}^{2D} \times \mathbb{R}^{2D} \to \mathbb{R}^{2D} \times \mathbb{R}^{2D}$, which outputs mean perturbations and per-dimension variances to calculate $\mu_{t|0,T}^{\bar{x}}, \mu_{t|0,T}^{\bar{v}}$ and $\Sigma_{t|0,T}^{\bar{x}}, \Sigma_{t|0,T}^{\bar{v}}$ and sample $(\bar{x}, \bar{v})$, as in (10). Note that we parameterize $\Sigma_{t|0,T}^{\bar{x}}, \Sigma_{t|0,T}^{\bar{v}}$ separately, matching the block diagonal form of (19). We calculate $v_{t|0,T}^{(q)}(\bar{x}, \bar{v}) := [v_{t|0,T}^{\bar{x}(q)}, v_{t|0,T}^{\bar{v}(q)}]$ from $u_{t|0,T}^{(q)}(\bar{x}, \bar{v}) = [u_{t|0,T}^{\bar{x}(q)}, u_{t|0,T}^{\bar{v}(q)}]$ as in (8)–(9), with $G_t^{-1} = (\frac{1}{2}\Xi_t\Xi_t^T)^{-1}$ given by (19). The Lagrangian objective in (6) minimizes the norm of the concatenated vector $v_{t|0,T}^{(q)}(\bar{x}, \bar{v})$, which depends on the reference drift $b_t(\bar{x}, \bar{v})$ in (19).

### B.2. Gaussian Mixture Paths

Note that the true Doob $h$-transform may not yield marginals which follow the unimodal Gaussian distributions in the previous sections. To increase the expressivity of our variational family of conditioned processes, we consider extending our parameterization to mixtures of Gaussians, $q_{t|0,T} \in \mathcal{Q}_{\text{MoG}}^K$. For a given set of $K$ mixture weights $w^k$ and component Gaussian paths $q_{t|0,T}^k$, the following identity allows us to recover the drift $u_{t|0,T}^{(q)}$ of the corresponding mixture distribution $q_{t|0,T}$.

**Proposition 4.** *Given a set of processes $q_{t|0,T}^k(x)$ and mixtures weights $w^k$, the vector field satisfying the Fokker-Planck equation in (7) for the mixture $q_{t|0,T}(x) = \sum_k w^k q_{t|0,T}^k(x)$ is given by*

$$u_{t|0,T}^{(q)}(x) = \sum_{k=1}^{K} \frac{w^k q_{t|0,T}^k(x)}{\sum_{j=1}^{K} w^j q_{t|0,T}^j(x)} u_{t|0,T}^{(q,k)}(x), \tag{20}$$

*where $u_{t|0,T}^{(q,k)}(x)$ satisfies the Fokker-Planck equation in (7) for $q_{t|0,T}^k(x)$. This identity holds for both first order dynamics in spatial coordinates only or second-order dynamics in $x = (\bar{x}, \bar{v})$.*

*Proof.* See Peluchetti (2023) Theorem 1 and its proof in their App. C. $\square$

Finally, we can calculate $v_{t|0,T}^{(q)}(x)$ by comparing $u_{t|0,T}^{(q)}(x)$ for the mixture of Gaussian path $q_{t|0,T} \in \mathcal{Q}_{\text{MoG}}^K$ to the reference drift $b_t(x)$ as in (9), and proceed to minimize its norm as in (6). We use Gumbel softmax reparamerization gradients (Maddison et al., 2016; Jang et al., 2017) to optimize the mixture weights $\{w^k\}_{k=1}^K$ alongside the neural network parameters $\{\theta^k\}_{k=1}^K$ for each Gaussian component $\{\mu_{t|0,T}^{(\theta)}, \Sigma_{t|0,T}^{(\theta)}\}_{k=1}^K$ and either first- or second-order dynamics.

## C. Proofs

### C.1. Proofs from Sec. 2.2 (Doob's $h$-Transform Background)

**Proposition. 2.**[Jamison (1975, Thm. 2)] *Let $h(x, t) := \rho(x_T \in \mathcal{B} \mid x_t = x)$ denote the conditional transition density with*

*respect to the reference process in* (3). *Letting* $G_t := \frac{1}{2}\Xi_t\Xi_t^T$, *the SDE*

$$dx_{t|0,T} = \Big(b_t(x_{t|0,T}) + 2G_t\nabla_x \log h(x_{t|0,T}, t)\Big) \cdot dt + \Xi_t\, dW_t \tag{21}$$

*is associated with the following transition probabilities*

$$\rho(x_t = y \,|\, x_s = x, x_T \in \mathcal{B}) = \frac{h(y,s)}{h(x,t)}\rho(x_t = y \,|\, x_s = x), \tag{22}$$

*for $s < t < T$, where we omit the dependence of $h(x,t)$ on $\mathcal{B}$ for simplicity of notation.*

*Proof.* See Jamison (1975) *for a simple proof based on Ito's Lemma, assuming smoothness and strict positivity of $h$.* □

**Proposition 3.** *The following PDEs are obeyed by (a) the marginal density of the conditioned process $\rho_{t|0,T}(x) = \rho(x_t = x \,|\, x_0 = A, x_T \in \mathcal{B})$ and (b) the h-function $h(x,t)$ (which implicitly depends on $\mathcal{B}$),*

$$\frac{\partial\rho_{t|0,T}(x)}{\partial t} + \Big\langle \nabla_x, \rho_{t|0,T}(x)\big(b_t(x) + 2G_t\nabla_x \log h(x,t)\big)\Big\rangle - \sum_{ij}(G_t)_{ij}\frac{\partial^2}{\partial x_i \partial x_j}\rho_{t|0,T}(x) = 0\,, \tag{12a}$$

$$\frac{\partial h(x,t)}{\partial t} + \Big\langle\nabla_x h(x,t), b_t(x)\Big\rangle + \sum_{ij}(G_t)_{ij}\frac{\partial^2}{\partial x_i \partial x_j}h(x,t) = 0\,. \tag{12b}$$

*Reparameterizing* (12b) *in terms of $s(x,t) := \log h(x,t)$, we can also write*

$$\frac{\partial s(x,t)}{\partial t} + \Big\langle\nabla s(x,t), G_t\nabla s(x,t)\Big\rangle + \Big\langle\nabla s(x,t), b_t(x_t)\Big\rangle + \sum_{ij}(G_t)_{ij}\frac{\partial^2}{\partial x_i \partial x_j}s(x,t) = 0. \tag{12c}$$

*Proof.* Let $p(x_{t+s} = y \,|\, x_t = x)$ denote the transition probability of a reference diffusion process

$$\frac{\partial}{\partial s}p(x_{t+s} = y \,|\, x_t = x) = -\Big\langle\nabla_y, p(x_{t+s} = y \,|\, x_t = x)b_{t+s}(y)\Big\rangle + \sum_{ij}(G_t)_{ij}\frac{\partial^2}{\partial y_i\partial y_j}p(x_{t+s} = y \,|\, x_t = x), \tag{23}$$

where $(G_t)_{ij} = \frac{1}{2}\Xi_{t+s}\Xi_{t+s}^T$.

Now we condition the process on the end-point value $x_T \in B$, and we get another kernel, i.e.

$$p(x_{t+s} = y \,|\, x_t = x, x_T \in B) = \frac{p(x_T \in B \,|\, x_{t+s} = y)}{p(x_T \in B \,|\, x_t = x)}p(x_{t+s} = y \,|\, x_t = x)\,, \tag{24}$$

We let $h(x,t) = p(x_T \in B \,|\, x_t = x)$ denote the conditional probability over the desired endpoint condition given $x_t = x$. According to laws of conditional probability, we can describe how $h(x,t)$ changes in time using the unconditioned transition probability

$$\underbrace{p(x_T \in B \,|\, x_t = x)}_{h(x,t)} = \int dy\ \underbrace{p(x_T \in B \,|\, x_{t+s} = y)}_{h(y,t+s)}p(x_{t+s} = y \,|\, x_t = x)\,, \tag{25}$$

we take the derivative $\frac{\partial}{\partial s}$ on both sides, and we get

$$0 = \int dy\ \left[p(x_{t+s} = y \,|\, x_t = x)\frac{\partial h(y, t+s)}{\partial s} + \frac{\partial p(x_{t+s} = y \,|\, x_t = x)}{\partial s}h(y, t+s)\right]. \tag{26}$$

Using the FP equation for the transition probability and integrating by parts, we have

$$0 = \int dy\, p(x_{t+s} = y \,|\, x_t = x)\left[\frac{\partial h(y, t+s)}{\partial s} + \Big\langle\nabla_y h(y, t+s), b_t(y)\Big\rangle + \sum_{ij}(G_t)_{ij}\frac{\partial^2}{\partial y_i\partial y_j}h(y, t+s)\right].$$

Note that this holds $\forall x$, hence, we have

$$\frac{\partial h(y, t+s)}{\partial s} + \left\langle \nabla_y h(y, t+s), b_{t+s}(y) \right\rangle + \sum_{ij} (G_t)_{ij} \frac{\partial^2}{\partial y_i \partial y_j} h(y, t+s) = 0\,,$$

without any loss of generality we can set $t = 0$

$$\frac{\partial h(y, s)}{\partial s} + \left\langle \nabla_y h(y, s), b_s(y) \right\rangle + \sum_{ij} (G_t)_{ij} \frac{\partial^2}{\partial y_i \partial y_j} h(y, s) = 0\,. \tag{27}$$

as desired to prove the optimality condition in (12b).

To prove (12a), denote $p(y, s) = p(x_s = y \mid x_0 = x)$ and differentiate $p(x_s = y \mid x_0 = x, x_T \in B) = \frac{h(y,s)}{h(x,0)} p(y, s)$ as

$$\begin{aligned}
&\frac{\partial}{\partial s} p(x_s = y \mid x_0 = x, x_T \in B) \\
&= \frac{1}{h(x, 0)} \left[ p(y, s) \frac{\partial h(y, s)}{\partial s} + h(y, s) \frac{\partial p(y, s)}{\partial s} \right] \\
&= \frac{1}{h(x, 0)} \left[ -\left\langle \nabla_y h(y, s), p(y, s) b_s(y) \right\rangle - p(y, s) \sum_{ij} (G_t)_{ij} \frac{\partial^2}{\partial y_i \partial y_j} h(y, s) \right. \\
&\qquad \left. - h(y, s) \left\langle \nabla_y, p(y, s) b_s(y) \right\rangle + h(y, s) \sum_{ij} (G_t)_{ij} \frac{\partial^2}{\partial y_i \partial y_j} p(y, s) \right] \\
&= -\left\langle \nabla_y, \frac{h(y, s)}{h(x, 0)} p(y, s) b_s(y) \right\rangle - p(y, s) \left\langle \nabla_y, 2D \nabla_y \frac{h(y, s)}{h(x, 0)} \right\rangle \\
&\qquad \pm \left\langle \nabla_y p(y, s), 2D \nabla_y \frac{h(y, s)}{h(x, 0)} \right\rangle + \sum_{ij} (G_t)_{ij} \frac{\partial^2}{\partial y_i \partial y_j} \left( \frac{h(y, s)}{h(x, 0)} p(y, s) \right)\,,
\end{aligned}$$

Note that $h(x, 0)$ can be pulled outside the differential operator because it is a function of $x$. The PDE for the new kernel $p(y, s \mid B) = p(x_s = y \mid x_0 = x, x_T \in B)$ (conditioned on the end-point) becomes

$$\frac{\partial}{\partial s} p(y, s \mid B) = -\left\langle \nabla_y, p(y, s \mid B)(b_s(y) + 2D \nabla_y \log h(y, s)) \right\rangle + \sum_{ij} (G_t)_{ij} \frac{\partial^2}{\partial y_i \partial y_j} p(y, s \mid B)\,. \tag{28}$$

which matches the desired PDE in (12a) thereby proving the first two parts of Prop. 3.

Finally, to show (12c), we index time using $t$ in Eq. (27) and change variables $h(x, t) = e^{s(x,t)}$,

$$\frac{\partial e^{s(x,t)}}{\partial t} + \left\langle \nabla_x e^{s(x,t)}, b_t(x) \right\rangle + \sum_{ij} (G_t)_{ij} \frac{\partial^2}{\partial x_i \partial x_j} e^{s(x,t)} = 0\,.$$

$$e^{s(x,t)} \frac{\partial s(x, t)}{\partial t} + e^{s(x,t)} \left\langle \nabla_x s(x, t), b_t(x) \right\rangle + \left\langle \nabla, G_t \nabla e^{s(x,t)} \right\rangle = 0$$

Next, we simplify $\left\langle \nabla, G_t \nabla e^{s(x,t)} \right\rangle = \left\langle \nabla, G_t e^{s(x,t)} \nabla s(x, t) \right\rangle = \left\langle \nabla e^{s(x,t)}, G_t \nabla s(x, t) \right\rangle + e^{s(x,t)} \left\langle \nabla, G_t \nabla s(x, t) \right\rangle = e^{s(x,t)} \left\langle \nabla s(x, t), G_t \nabla s(x, t) \right\rangle + e^{s(x,t)} \left\langle \nabla, G_t \nabla s(x, t) \right\rangle$ to finally write

$$e^{s(x,t)} \left( \frac{\partial s(x, t)}{\partial t} + \left\langle \nabla_x s(x, t), b_t(x) \right\rangle + \left\langle \nabla s(x, t), G_t \nabla s(x, t) \right\rangle + \sum_{ij} (G_t)_{ij} \frac{\partial^2}{\partial x_i \partial x_j} s(x, t) \right) = 0$$

which demonstrates (12c) since the inner term must be zero. $\qquad \square$

**C.2. Proofs from Sec. 3.1 (Lagrangian Action Minimization for Doob's $h$-Transform)**

We begin by proving Cor. 1, whose proof actually contains the initial steps needed to prove our main theorem Thm. 1. In both proofs, we omit conditioning notation $q_t \leftarrow q_{t|0,T}$ for simplicity and assume $q_t(x)s_t(x) \to 0$ vanishes at the boundary $x \to \pm\infty$, which is used when integrating by parts in $x$.

**Corollary 1.** *The Lagrangian objective in Thm. 1 which solves Doob's h-transform is equivalent to*

$$S = \min_q \max_s \ s(B,1) - s(A,0) - \int_0^1 dt \int dx \ q_{t|0,T}\left(\frac{\partial s}{\partial t} + \left\langle \nabla s, G_t \nabla s \right\rangle + \left\langle \nabla s, b_t \right\rangle + \left\langle \nabla, G_t \nabla s \right\rangle\right)$$

*if $q_{t|0,T}$ satisfies (6c). Note $v_{t|0,T}(x) = \nabla_x s(x,t)$, with $s^*(x,t) = \log h(x,t)$ at optimality.*[3]

*Proof.* Consider the following action functional

$$
\begin{aligned}
S = \ &\min_{q,v} \int dt \int dx \ q_t(x)\langle v_t(x), G_t v_t(x)\rangle, \\
&\text{s.t.} \ \ \frac{\partial q_t(x)}{\partial t} = -\left\langle \nabla_x, q_t(x)(b_t(x) + 2G_t v_t(x))\right\rangle + \sum_{ij}(G_t)_{ij}\frac{\partial^2}{\partial x_i \partial x_j}q_t(x), \\
&\quad\ \ q_0(x) = \delta(x - A), \ \ q_1(x) = \delta(x - B).
\end{aligned}
$$

The Lagrangian of this optimization problem is

$$\mathcal{L} = \int_0^1 dt \int dx \left[ q_t\langle v_t, G_t v_t\rangle + s_t\left(\frac{\partial q_t}{\partial t} + \left\langle \nabla, q_t(b_t + 2G_t v_t)\right\rangle - \sum_{ij}(G_t)_{ij}\frac{\partial^2}{\partial x_i \partial x_j}q_t\right)\right],$$

where $s_t$ is the dual variable and we omit the optimization arguments, with $S = \min_{q,v}\max_s \mathcal{L}$. Swapping the order of optimizations under strong duality, we take the variation with respect to $v_t$ in an arbitrary direction $\mathfrak{h}_t$. Using $G_t = G_t^T$, we obtain

$$
\begin{aligned}
\frac{\delta\mathcal{L}}{\delta v_t}[\mathfrak{h}_t] = \ &q_t\left\langle (G_t + G_t^T)v_t, \mathfrak{h}_t\right\rangle - q_t\left\langle 2G_t^T\nabla s_t, \mathfrak{h}_t\right\rangle = 0 \\
&\implies v_t = \nabla s_t,
\end{aligned}
\tag{29}
$$

Substituting into the above, we have

$$\mathcal{L} = \int_0^1 dt \int dx \left[ s_t\frac{\partial q_t}{\partial t} - q_t\left\langle \nabla s_t, G_t\nabla s_t\right\rangle + s_t\left\langle \nabla, q_t b_t\right\rangle - s_t\left\langle \nabla, G_t\nabla q_t\right\rangle\right]. \tag{30}$$

Integrating by parts in $t$ and in $x$, assuming that $q_t(x)s_t(x) \to 0$ as $x \to \pm\infty$, yields

$$
\begin{aligned}
\mathcal{L} &= \int dx \ q_1 s_1 - \int dx \ q_0 s_0 + \int_0^1 dt \int dx \left[-q_t\frac{\partial s_t}{\partial t} - q_t\left\langle \nabla s_t, G_t\nabla s_t\right\rangle - q_t\left\langle \nabla s_t, b_t\right\rangle + \left\langle \nabla s_t, G_t\nabla q_t\right\rangle\right] \\
&= \int dx \ q_1 s_1 - \int dx \ q_0 s_0 + \int_0^1 dt \int dx \left[-q_t\frac{\partial s_t}{\partial t} - q_t\left\langle \nabla s_t, G_t\nabla s_t\right\rangle - q_t\left\langle \nabla s_t, b_t\right\rangle - q_t\left\langle \nabla, G_t\nabla s_t\right\rangle\right] \\
&= \int dx \ q_1 s_1 - \int dx \ q_0 s_0 - \int_0^1 dt \int dx \ q_t\left[\frac{\partial s_t}{\partial t} + \left\langle \nabla s_t, G_t\nabla s_t\right\rangle + \left\langle \nabla s_t, b_t\right\rangle + \left\langle \nabla, G_t\nabla s_t\right\rangle\right]
\end{aligned}
\tag{31}
$$

where in the second line, we integrate by parts in $x$ again. Enforcing $q_1(x) = \delta(x - B)$ and $q_0(x) = \delta(x - A)$ and recalling $S = \min_q \max_s \mathcal{L}$ after eliminating $v_t$, we recover the optimization in the statement of the corollary. $\qquad\square$

_______________

[3]Again, note that we omit the dependence of $s(x,t)$ and $h(x,t)$ on the conditioning information $B$.

**Theorem. 1.** *The following Lagrangian action functional has a unique solution which matches the Doob $h$-transform in* *Prop. 3,*

$$\mathcal{S} = \min_{q,v} \int_0^T dt \int dx \, q_{t|0,T}(x)\langle v_{t|0,T}(x), G_t \, v_{t|0,T}(x)\rangle, \tag{32a}$$

$$s.t. \quad \frac{\partial q_{t|0,T}(x)}{\partial t} = -\langle \nabla_x, q_{t|0,T}(x)\big(b_t(x) + 2G_t \, v_{t|0,T}(x)\big)\rangle + \sum_{ij}(G_t)_{ij}\frac{\partial^2}{\partial x_i \partial x_j}q_{t|0,T}(x), \tag{32b}$$

$$q_0(x) = \delta(x - A), \qquad q_T(x) = \delta(x - B). \tag{32c}$$

*Namely, the optimal $q^*_{t|0,T}(x)$ obeys (12a) and the optimal $v^*_{t|0,T}(x) = \nabla_x \log h(x,t) = \nabla_x s(x,t)$ follows (12b) or (12c).*

*Proof.* The proof proceeds from (30) above,

$$\mathcal{S} = \min_q \max_s \mathcal{L} = \min_q \max_s \int_0^1 dt \int dx \left[ s_t \frac{\partial q}{\partial t} - q_t\langle \nabla s_t, G_t \nabla s_t\rangle + s_t\langle \nabla, q_t b_t\rangle - s_t\langle \nabla, G_t \nabla q_t\rangle \right]. \tag{33}$$

We first show that the optimality condition with respect to $s_t$ yields the Fokker-Planck equation for $q_t$ in Prop. 3 (12a), before deriving the PDE in (12b) as the optimality condition with respect to $q_t$.

*Optimality Condition for (32) recovers Prop. 3 (12a):* The variation with respect to $s_t$ of (33) is simple, apart from the intermediate term. For a perturbation direction $\mathfrak{h}_t$, we seek

$$\int dx \, \frac{\delta(\cdot)}{\delta s_t}\mathfrak{h}_t = \frac{d}{d\varepsilon}\left[ -\int dx \, q_t\langle \nabla(s_t + \varepsilon\mathfrak{h}_t), G_t \nabla(s_t + \varepsilon\mathfrak{h}_t)\rangle \right]\Big|_{\varepsilon=0},$$

where $\cdot$ indicates the functional on the right hand side. Proceeding to differentiate with respect to $\varepsilon$, we use linearity to pull $\frac{d}{d\varepsilon}$ inside the integral and apply it first to obtain $\frac{d}{d\varepsilon}(s_t + \varepsilon\mathfrak{h}_t) = \mathfrak{h}_t$. Using the product rule, recognizing the symmetry of terms, and evaluating at $\varepsilon = 0$, we are left with

$$\int dx \, \frac{\delta(\cdot)}{\delta s_t}\mathfrak{h}_t = \left[ -2\int dx \, q_t\langle \nabla\mathfrak{h}_t, G_t \nabla s_t\rangle \right] \overset{(i)}{=} \left[ \int dx \, \mathfrak{h}_t\big(2\langle \nabla, q_t G_t \nabla s_t\rangle\big) \right] \tag{34}$$

where in $(i)$ we integrate by parts $x$.

We are now ready to set the variation of (33) with respect to $s_t$ (in an arbitrary direction $\mathfrak{h}_t$) equal to zero. Using (34), we have

$$\frac{\delta\mathcal{L}}{\delta s_t}[\mathfrak{h}_t] = 0 = \frac{\partial q_t}{\partial t} + 2\langle \nabla, q_t G_t \nabla s_t\rangle + \langle \nabla, q_t b_t\rangle - \langle \nabla, G_t \nabla q_t\rangle$$

$$\implies \quad 0 = \frac{\partial q_t}{\partial t} + \langle \nabla, q_t\big(b_t + 2G_t \nabla s_t\big)\rangle - \langle \nabla, G_t \nabla q_t\rangle \tag{35}$$

which matches the desired optimality condition for the conditioned process in Prop. 3 (12a).

*Optimality Condition for (32) recovers Prop. 3 (12b):* Starting again from (33), we take the variation with respect to $q_t$. First, we repeat identical steps (integrate by parts in both $x$ and $t$) to reach (31),

$$\mathcal{L} = \int dx \, q_1 s_1 - \int dx \, q_0 s_0 - \int_0^1 dt \int dx \, q_t\left[ \frac{\partial s_t}{\partial t} + \langle \nabla s_t, G_t \nabla s_t\rangle + \langle \nabla s_t, b_t\rangle + \langle \nabla, G_t \nabla s_t\rangle \right]$$

where it is now clear that taking the variation with respect to $q_t$ and setting equal to zero yields

$$\frac{\delta\mathcal{L}}{\delta q_t}[\mathfrak{h}_t] = 0 = \frac{\partial s_t}{\partial t} + \langle \nabla s_t, G_t \nabla s_t\rangle + \langle \nabla s_t, b_t\rangle + \langle \nabla, G_t \nabla s_t\rangle \tag{36}$$

which is the desired PDE for $s(x,t) = \log h(x,t)$ in (12c). To obtain (12b), we note an identity used to simplify the last term

$$\sum_{ij}(G_t)_{ij}\frac{\partial^2}{\partial x_i \partial x_j}\log h_t = \langle \nabla, G_t \nabla \log h_t\rangle = \left\langle \nabla, \frac{1}{h_t}G_t \nabla h_t\right\rangle = -\frac{1}{h_t^2}\langle \nabla h_t, G_t \nabla h_t\rangle + \frac{1}{h_t}\langle \nabla, G_t \nabla h_t\rangle.$$

Now, substituting $s(x, t) = \log h(x, t)$ into Eq. (36), we obtain

$$\frac{1}{h_t}\frac{\partial h_t}{\partial t} + \frac{1}{h_t^2}\langle\nabla h_t, G_t\nabla h_t\rangle + \frac{1}{h_t}\langle\nabla h_t, b_t\rangle - \frac{1}{h_t^2}\langle\nabla h_t, G_t\nabla h_t\rangle + \frac{1}{h_t}\langle\nabla, G_t\nabla h_t\rangle = 0\,,$$

$$\implies \quad \frac{\partial h_t(x)}{\partial t} + \langle\nabla h_t(x), b_t(x)\rangle + \langle\nabla, G_t\nabla h_t\rangle = 0, \tag{37}$$

which matches (12b) as desired.

The last equation defines the backward Kolmogorov equation for the diffusion process with the drift $b_t(x)$ and covariance matrix $G_t$, i.e. the function $h_t(x)$ defines the conditional density $h_t(x) = p(x_T \in B' \mid x_t = x)$ for some set $B'$, which agrees with the forward process with the same drift and covariance. The boundary condition $q_T(x) = \delta(x - B)$ together with the backward equation define the unique solution to this PDE. Since the PDEs and the boundary conditions are the same as in Doob's $h$-transform, we have $h_t(x) = p(x_T = B \mid x_t = x)$. $\qquad\square$

**Corollary 2.** *The Lagrangian objective in Thm. 1 is equivalent to the following optimization of $\mathbb{Q}_{0:T}^v$*

$$\mathcal{S} := \min_{\mathbb{Q} \ s.t. \ \mathbb{Q}=\delta, \mathbb{Q}=\delta} D_{KL}[\mathbb{Q}_{0:T}^v : \mathbb{P}_{0:T}^{ref}] \tag{14}$$

*where the minimizing argument recovers the path measure $\mathbb{P}_{0:T}^*$ associated with the SDE in (4).*

*Proof.* We use the Girsanov theorem (Särkkä and Solin, 2019, Sec. 7.3) to calculate the KL divergence between the following two Brownian diffusions with fixed initial condition $x_0 = A$,

$$\mathbb{P}_{0:T}^{ref}: \qquad dx_t = b_t(x_t) \cdot dt + \Xi_t\, dW_t\,, \tag{38}$$

$$\mathbb{Q}_{0:T}^v: \qquad dx_t = \big(b_t(x_t) + 2G_t\, v_{t|0,T}(x_{t|0,T})\big) \cdot dt + \Xi_t\, dW_t\,, \tag{39}$$

In particular, noting the difference of drifts is $b_t(x_t) + 2G_t\, v_{t|0,T}(x_t) - b_t(x_t) = 2G_t\, v_{t|0,T}(x_t)$, the likelihood ratio is given by

$$\frac{d\mathbb{Q}_{0:T}^v}{d\mathbb{P}_{0:T}^{ref}} = \frac{q_{t|0,T}(x_0, ...x_T)}{\rho(x_0, ...x_T)} = \exp\Big\{-\frac{1}{2}\int_0^T\langle 2G_t\, v_{t|0,T}(x_t), (G_t)^{-1}\, 2G_t\, v_{t|0,T}(x_t)\rangle dt \tag{40}$$

$$-\int 2\big(G_t\, v_{t|0,T}(x_t)\big)^T G_t^{-1}dW_t\Big\}$$

We finally calculate the KL divergence, noting that, after taking the log, the expectation of the integral $\int(\cdot)dW_t$ in the final term vanishes,

$$D_{KL}[\mathbb{Q}_{0:T}^v : \mathbb{P}_{0:T}^{ref}] = 2\int_0^1 dt\,\int dx_t\, q_{t|0,T}(x_t)\,\langle v_{t|0,T}(x_t), G_t\, v_{t|0,T}(x_t)\rangle, \tag{41}$$

which matches (6a) up to a constant factor of 2 does not change the optimum. We finally compare to the constraints in Thm. 1. First, it is clear that the diffusion in (39) satisfies the Fokker-Planck equation in (6b) (Särkkä and Solin, 2019, Sec. 5.2). We respect (6c) by optimizing over endpoint-constrained path measures, which yields

$$\mathcal{S} = \min_{\mathbb{Q} \ s.t. \ \mathbb{Q}=\delta, \mathbb{Q}=\delta} D_{KL}[\mathbb{Q}_{0:T}^v : \mathbb{P}_{0:T}^{ref}] \tag{42}$$

as desired. $\qquad\square$

# D. Related Work

**(Aligned) Schrödinger Bridge Matching Methods.** Many existing 'bridge matching' approaches (Shi et al., 2023; Peluchetti, 2021; 2023; Liu et al., 2022; Lipman et al., 2022; Liu et al., 2023b) for SB and generative modeling rely on convenient properties of Brownian bridges and would require calculating $h$-transforms to simulate bridges for general reference processes. Our conditional Gaussian path parameterization is similar to Liu et al. (2023a); Neklyudov et al. (2024), where analytic bridges are not available for SB problems with nonlinear reference drift or general costs.

Somnath et al. (2023); Liu et al. (2023b) attempt to solve the SB problem given access to aligned data $x_0, x_T \sim q_{0,T}^{\text{data}}$ assumed to be drawn from an optimal coupling. While the method in Somnath et al. (2023) involves approximating an $h$-transform, their goal is to obtain an unconditioned vector field $v_t$ to simulate a Markov process. However, De Bortoli et al. (2023) use Doob's $h$-transform to argue the learned Markov process will not preserve the empirical coupling unless $q_{0,T}^{\text{data}}$ is the optimal coupling for the SB problem, and show that an 'augmented' $v_{0,t}$ which conditions on $x_0$ can correct this issue.

After training on a dataset of $x_0, x_T \sim q_{0,T}^{\text{data}}$ pairs using our method, we could consider using an (augmented) bridge matching objective (Shi et al., 2023; De Bortoli et al., 2023) to distill our learned $v_{t|0,T}^{(q)}$ into a vector field $v_t$ or $v_{0,t}$ which does not condition on the endpoint. Our use of a Gaussian path parameterization with samples from a fixed endpoint coupling and no Markovization step corresponds to a simplified version of the conditional optimal control step in Liu et al. (2023a).

**Transition Path Sampling.** We refer to the surveys of Dellago et al. (2002); Weinan and Vanden-Eijnden (2010); Bolhuis and Swenson (2021) for an overview of the TPS problem. Least action principles for TPS have a long history, building upon the Freidlin-Wentzell (Freidlin and Wentzell, 1998) and Onsager-Machlup (Onsager and Machlup, 1953; Dürr and Bach, 1978) Lagrangian functionals in the zero-noise limit and finite-noise cases. In particular, the Onsager-Machlup functional relates maximum a posteriori estimators or 'most probable (conditioned) paths' to the minimizers of an action functional similar to Thm. 1, where example algorithms include (Vanden-Eijnden and Heymann, 2008; Sheppard et al., 2008). By contrast, our approach targets the *entire* posterior over transition paths using an expressive variational family. While Lu et al. (2017) provide analysis for the Gaussian family, we draw connections with Doob's $h$-transform and extend to mixtures of Gaussians.

Shooting methods are among the most popular for sampling the posterior of transition paths. From a path that satisfies the boundary conditions (obtained, e.g., using high-temperature simulations), shooting picks points and directions in which to propose alterations, then simulates new trajectories and accepts or rejects using Metropolis-Hastings (MH) (Juraszek and Bolhuis, 2008; Borrero and Dellago, 2016; Jung et al., 2017; Falkner et al., 2023; Jung et al., 2023). While the MCMC corrections yield theoretical guarantees, shooting methods involve expensive molecular dynamics (MD) simulations and need to balance high rejection rates with large changes in trajectories. One-way shooting methods sample paths efficiently but yield highly correlated samples. Two-way shooting methods, which we compare against in Sec. 4, are more expensive but typically sample diverse paths faster. Recent machine learning approaches such as Plainer et al. (2023); Lelièvre et al. (2023) aim to reduce the need for MD. Holdijk et al. (2024) propose a stochastic optimal control method that simulates (13) with a learned drift, but can be inefficient unless the terminal state is sampled frequently.

**Machine Learning for Molecular Simulation.**

The main dilemma of molecular dynamics comes from the accuracy and efficiency trade-off—accurate simulation requires solving the Schrödinger equation which is computationally intractable for large systems, while efficient simulation relies on empirical force fields which is inaccurate. Recently, there has been a surge of work in applying machine learning approaches to accelerate molecular simulation. One successful paradigm is machine learning force field (MLFF) which leverages the transferability and efficiency of machine learning methods to fit force/energy prediction models on quantum mechanical data and transfer across different atomic systems (Smith et al., 2017; Wang et al., 2018). More recently, increasing attention has been focused on building atomic foundation models to encompass all types of molecular structures (Batatia et al., 2023; Shoghi et al., 2023; Zhang et al., 2022).

Sampling is a classical problem in molecular dynamics to draw samples from the Boltzmann distribution of molecular systems. Classical methods mainly rely on Markov chain Monte Carlo (MCMC) or MD which requires long mixing time for multimodal distributions with high energy barriers (Rotskoff, 2024). Generative models in machine learning demonstrate promises in alleviating this problem by learning to draw independent samples from the Boltzmann distribution of molecular systems (known as Boltzmann generator) (Noé et al., 2019). Numerous methods have been developed to utilize generative models as a proposal distribution for escaping local minima in running MCMC methods (Gabrié et al., 2022). However, one critical issue is that generative models rely on training from samples. Although recent advances have been developed to learn from unnormalized density (i.e., energy) function, the training inefficiency limits their applicability to solve high-dimensional molecular dynamics problems. To circumvent the curse of dimensionality for the sampling problem, another branch of work study to learn coarse-grained representation with neural networks (Sidky et al., 2020). For broader literature of applying machine learning to enhanced sampling, we refer the reader to Mehdi et al. (2024).

# E. Further Experimental Details

## E.1. Evaluation Metrics

To assess the quality of our approach in terms of performance and physicalness of paths, we compare them under different metrics to well-established TPS techniques. One important describing factor of a trajectory is the molecule's highest energy during the transition. These high-energy states are often referred as transition states and less likely to occur but they determine importance factors during chemical reaction such as reaction rate. As such, we will look at the maximum energy along the transition path and use it to compare the ensemble of trajectories more efficiently. The main goal is to show that lower energy of the transition states can be sampled by the methods.

However, the maximum energy does not account for the fact that the transition path needs to be sequential, and each step needs to be coherent based on the previous position and momentum. For this, we also compare the likelihood of the paths (i.e., unnormalized density) by computing the probably of being in the start state $\rho(x_0)$ and multiplying it with the step probability such that

$$L(x_0, x_1, \ldots, x_{N-1}) = \rho(x_0) \cdot \prod_{i=0}^{N-2} \pi(x_{i+1} \mid x_i). \tag{43}$$

For the step probability $\pi$, we solve the Langevin leap-frog implementation as implemented in OpenMM to solve $\mathcal{N}(x_{i+1} \mid x_i + dt \cdot b_t(x), dt\sigma_i^2)$. As for the starting probability, we compute the unnormalized density of the Boltzmann distribution for our start state $z$ and assume that the velocity $v$ can be sampled independently (Castellan, 1983, Sec. 4.6)

$$\rho(z, v) \propto \exp\left(-\frac{U(z)}{k_B T}\right) \cdot \mathcal{N}\left(v \mid 0, k_B T \cdot M^{-1}\right), \tag{44}$$

with the Boltzmann constant $k_B$ and the diagonal matrix $M$ containing the mass of each atom.

As for the performance, the number of energy evaluations will be the main determining factor of the runtime for larger molecular systems, especially for proteins. We hence compare the use of the number of energy computations as a proxy for hardware-independent relative measurements. In our tests, this number aligned with the relative runtime of these approaches.

## E.2. Toy Potentials

The toy systems move according to the following integration scheme (first-order Euler)

$$x_{t+1} = x_t - dt \cdot \nabla_x U(x_t) + \sqrt{dt} \cdot \text{diag}(\xi) \cdot \varepsilon, \quad \varepsilon \sim \mathcal{N}(0, 1), \tag{45}$$

following the definition of our stochastic system in Sec. 2.2 with a time-independent Wiener process, where $\xi$ is a constant time-independent standard deviation for all dimensions.

**Müller-Brown.** The underlying Müller-Brown potential that has been used for our experiments can be written as

$$\begin{aligned} U(x, y) = &- 200 \cdot \exp\left(-(x-1)^2 - 10y^2\right) \\ &- 100 \cdot \exp\left(-x^2 - 10 \cdot (y-0.5)^2\right) \\ &- 170 \cdot \exp\left(-6.5 \cdot (0.5 + x)^2 + 11 \cdot (x+0.5) \cdot (y-1.5) - 6.5 \cdot (y-1.5)^2\right) \\ &+ 15 \cdot \exp\left(0.7 \cdot (1+x)^2 + 0.6 \cdot (x+1) \cdot (y-1) + 0.7 \cdot (y-1)^2\right). \end{aligned} \tag{46}$$

We used a first-order Euler integration scheme to simulate transition paths with 275 steps and a $dt$ of $10^{-4}s$. $\xi$ was chosen to be 5 and 1,000 transition paths were simulated. We have used an MLP with four layers and a hidden dimension of 128 each, with swish activations. It has been trained for 2,500 steps with a batch size of 512.

In Fig. 5a, we compare the likelihood of the sampled paths. We can see that one-way shooting takes time until the path is decorrelated from the initial trajectory, which is shorter and thus has a higher likelihood. All MCMC methods exhibit this behavior, which is typically alleviated by using a warmup period in which all paths are discarded. After that, all methods exhibit similar likelihood, with our method having a slightly lower likelihood. Looking at the maximum energy on the trajectory in Fig. 5b reveals that all methods have a similar quality of paths.

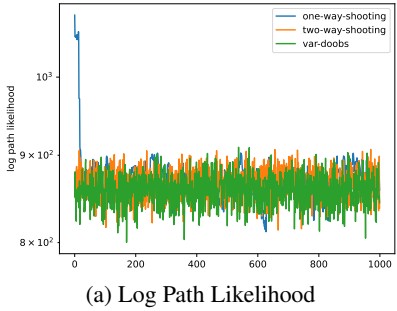
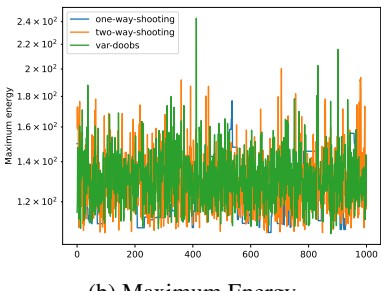

(a) Log Path Likelihood                              (b) Maximum Energy

Figure 5: In a, we compare the log likelihood of sampled trajectories, where a higher likelihood is generally more favorable. The plot in b shows the maximum energy of each individual trajectory. A high maximum energy means that the molecule needs to be in an excited state during the transition, making it less likely to occur under lower temperatures.

**Dual-Channel Double-Well.** To demonstrate the advantage of mixtures, we have used the two-dimensional potential

$$
\begin{aligned}
U(x,y) = &+ 2 \cdot \exp\big(-(12x^2 + 12y^2)\big) \\
&- 1 \cdot \exp\big(-(12 \cdot (x+0.5)^2 + 12y^2)\big) \\
&- 1 \cdot \exp\big(-(12 \cdot (x-0.5)^2 + 12y^2)\big) + x^6 + y^6 \, .
\end{aligned}
\tag{47}
$$

In this case, we have used $dt = 5 * 10^{-4}s$ with a transition time of $T = 1s$ and $\xi = 0.1$. As for the MLP, we have used the same structure as in the Müller-Brown example but trained it for 20,000 iterations. The corresponding weights to Prop. 4 are $w = [\frac{1}{2}, \frac{1}{2}]$ and are fixed for this experiment and hence $w \notin \theta$.

### E.3. Neural Network Parameterization

We parameterize our model with neural networks, a 5-layer MLP with ReLU activation function and 256/512 hidden units for alanine dipeptide and Chignolin, respectively. The neural networks are trained using an Adam optimizer with learning rate $10^{-4}$.

We represent the molecular system in two ways: (1) in Cartesian coordinates, which are the 3D coordinates of each atoms, and with (2) internal coordinate which instead uses bond length, angle and dihedral angle along the molecule, where we use the same parameterization as in (Noé et al., 2019).

Our state definition includes a variance parameter for the initial and target marginal distributions at $t = 0$ and $t = T$, we choose the variance to be $10^{-8}$ which almost does not change the energy of the perturbed system.

### E.4. Molecular Systems

To simulate molecular dynamics, we rely on the AMBER14 forcefield (amber14/protein.ff14SB (Maier et al., 2015)) without a solvent, as implemented in OpenMM (Eastman et al., 2017). As OpenMM does not support auto-differentiation, we do not use OpenMM for the simulations themselves, but utilize DMFF (Wang et al., 2023) which is a differentiable framework implemented in JAX (Bradbury et al., 2018) for molecular simulation. This is needed because during training we compute $\nabla_\theta U\big(x_{t|0,T} \sim \mathcal{N}(\mu_{t|0,T}^{(\theta)}, \Sigma_{t|0,T}^{(\theta)})\big)$, where the concrete $x_{t|0,T}$ is sampled based on the parameters of the neural network.

For the concrete simulations, we ran them with the timestep $dt = 1fs$, with $T = 1ps$, $\gamma = 1ps$, and Temp $= 300K$. To compute the MCMC two-way shooting baselines, we use the same settings and consider trajectories as failed, if they exceed 2,000 steps without reaching the target.

**Visualization of transition for alanine dipeptide.** In Fig. 6, we show a transition sampled without any noise from the model with internal coordinates and 2 Gaussian mixtures.

### E.5. Computational Resources

All our experiments involving training were conducted on a single NVIDIA A100 80GB. The baselines themselves were computed on a M3 Pro 12-core CPU.

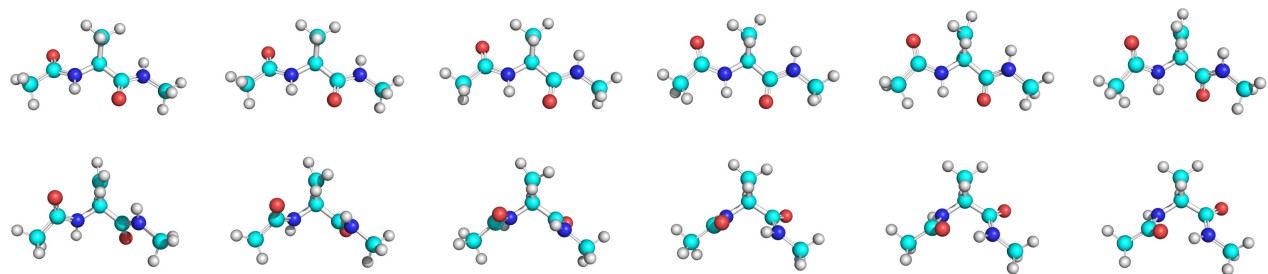

Figure 6: Transition path for the alanine dipeptide.

## F. Societal Impact

Our research concerns the efficient sampling of transition paths which are crucial for a variety of tasks in biology, chemistry, materials science and engineering. Our research could potentially benefit research areas from combustion, catalysis, protein design to battery design. Nevertheless, we do not foresee special potential negative impacts to be discussed here.