# OpenReview forum: "Doob's Lagrangian: A Sample-Efficient Variational Approach to Transition Path Sampling"
_ICML.cc/2024/Workshop/ML4LMS — ML4LMS Poster_

### Official Review · Reviewer_56vz · 2024-06-12
**Doob's Lagrangian**

**Rating:** 8
**Confidence:** 4

**Review:**

could use more comparisons to many more existing approaches.

---

### Official Review · Reviewer_7bsc · 2024-06-12
**Interesting paper with theory and application. Theory part could benefit from minor changes helping clarity.**

**Rating:** 8
**Confidence:** 3

**Review:**

The key idea in the paper is to learn a Gaussian distribution for $q_{t|0,T}(x)$, the mean and covariance of which would be given by a neural net taking t, the initial point $x_0=A$ and the final point $x_T=B$ as an input. There is a variational objective function which is used to train the neural net. For transition paths in large deviation theory, we expect Gaussian deviations from the optimal path, so this approach makes sense. The method is applied to transitions in some artificial potential landscapes as well as to a small peptide and a protein.

As to complaints,
1) Figure 1 is too small to read. I know that the two-column format is a challenge but
2) The Doob $h$-transform is defined in subsection 2.2 for a set $\cal B$. By the time one sees Theorem 1, it is not clear why $q_T(x)=\delta(x-B)$. Unless one looks ahead to subsection 3.1, it is not clear that the authors have specialized the case where ${\cal B}=\{ B \}$. Going to the proof in Appendix C1, prop 3 only confuses, since the proof switches from $\cal B$ to $B$ halfway where $B$ still signifies a set and not a single point.

---

### Official Review · Reviewer_CvtL · 2024-06-12

**Rating:** 7
**Confidence:** 2

**Review:**

In this work, the authors proposed a variational formulation of Doob's $h$-transformation for the purpose of efficient rare event sampling. The claimed major advantage of such method is sample-efficiency and simulation-free. From my perspective, the proposed method share synergy with the flow-matching method, while being trained on a VAE-like objective. After being trained, the model is capable of sample state between starting point $A$ and end point $B$ for any given timestep t in one shot. Overall, it looks like it is a novel idea and can bring research attention to rare event sampling with variational methods. However, I do think that the authors should provide more details on how the model is trained: details of loss function, training data pair, etc. Current description of method is vague and requires the reader to have strong prior knowledge of both Doob's $h$-transformation, flow-based models, and variational methods. Also, some part of the claims are not clear. For example, the authors mention that the $t(1-t)$ coefficient in eqn 10 helps to address challenge 3, but does not provide detailed explanation of such claim.

---

### Official Review · Reviewer_vyDk · 2024-06-12

**Rating:** 7
**Confidence:** 3

**Review:**

**Summary:**

The authors propose a method to sample transition paths between two known states, based on a Lagrangian that has Doob's h-transform as a solution and an efficient parametrization that is analytically tractable. The specific parameters for a given problem are then approximated through a simple multilayer perceptron.

**Pros:**

The method is novel and proposes to solve a hard problem with important applications.
The method seems to work well on the systems shown, that range from toy models to the limit of what is nowadays tractable.

**Cons:**

While the number of potential evaluations is a relevant measure for the cost of an approach, specific methods can have different overheads. In particular, the present method needs to train a neural network on a certain number of samples, where a more expensive gradient calculation is required. Each evaluation then requires at least an inference step for the network, which can still be a considerable cost with respect to the simplest potentials. I think it would be more fair to report the total computational cost for training and sampling for each method.

For structured problems, it would be interesting to see some comments on the possibility to use a more advanced neural network architecture that takes into account the physics of the system, similar to the machine learned interatomic potentials mentioned in the appendix.

**Other remarks:**

I might be confused on the notation, but shouldn't the fraction in Eq.5 read $h(y,t)/h(x,s)$?

---

### Official Review · Reviewer_wy4x · 2024-06-12

**Rating:** 4
**Confidence:** 3

**Review:**

This paper presented an alternative way to obtain the probable transition path between two metastable states. The problem of interest is relevant and important and often approached by MCMC, MD and/or enhanced sampling methods, which yet can be computationally extensive. This work proposed a variational formulation for the task of finding transition paths as opposed to traditional simulation. However, even the paper is theory-abundant, very few fresh insights can be revealed with the given baseline and experimental settings. Some related works such as enhanced sampling for rare events from the recent years can be included for comparison. Also, there still exists some space to improve the presentation.

---

### Official Review · Reviewer_yay8 · 2024-06-12
**Rigorous math, can add more details**

**Rating:** 9
**Confidence:** 4

**Review:**

This is a paper about rare event sampling. In the particular case of Brownian motion, Doob's h-transform can be used to model the conditioning process, but simulation of h-transform is difficult due to the large trajectories search space. This paper proposed how to avoid such expensive simulation.

The formulation of Doob's h-transform is in sec 2.2, but using it directly would be difficult. The authors reshaped the h-transform into a constrained minimization problem (theorem 1) and in turn, proposed a parameterization technique to avoid sampling by simulation, and introduce analytic solutions to constraints. The derivation is long, at the appendix of the paper, and the summary is at sec.3. The key contribution of this paper is at equation (10), which instead of finding the integrals of equation (6a) by Monte Carlo sampling, it is solved by using a neural network, in which the neural network is considered as an approximator to the integral. This avoids simulation to find a sample path.

The math derivation is long, which makes this paper difficult to read. The result is as strong as sec.4 shows. However, little is mentioned about the neural network (the best is a few lines in appendix E.3). This is the major limitation to allow others to benefit from the paper. In particular, the authors have not explore the different design space of neural networks and how much impact to the conclusion. Further, how the training/validation data to the neural network are prepared (which I suspect can be significant to the accuracy but not on efficiency) is also not addressed.